# Orca: A Few-shot Benchmark for Chinese Conversational Machine Reading Comprehension

**Nuo Chen[1]∗, Hongguang Li[2], Junqing He[3], Yinan Bao[4], Xinshi Lin[2], Qi Yang[3],
Jianfeng Liu[2], Ruyi Gan[3], Jiaxing Zhang[3], Baoyuan Wang[2]†, Jia Li[1]†**

[1] Hong Kong University of Science and Technology (Guangzhou),
Hong Kong University of Science and Technology

[2]Xiaobing.AI

[3]CCNL, IDEA, Shenzhen, China

[4] Institute of Information Engineering, Chinese Academy of Sciences

## Abstract

The Conversational Machine Reading Comprehension (CMRC) task aims to answer questions in conversations, which has been a hot research topic because of its wide applications. However, existing CMRC benchmarks in which each conversation is coupled with a static passage are inconsistent with real scenarios. In this regard, it is hard to evaluate model's comprehension ability towards real scenarios. In this work, we propose the first Chinese CMRC benchmark **Orca** and further provide zero-shot/few-shot settings to evaluate model's generalization ability towards diverse domains. We collect 831 hot-topic driven conversations with 4,742 turns in total. Each turn of a conversation is assigned with a response-related passage, aiming to evaluate model's comprehension ability more reasonably. The topics of conversations are collected from social media platform and cover 33 domains, trying to be consistent with real scenarios. Importantly, answers in Orca are all well-annotated natural responses rather than specific spans or short phrases in previous datasets. We implement two strong frameworks to tackle the challenge in Orca. The results indicate there is substantial room for improvement for strong baselines such as ChatGPT on our CMRC benchmark. Our codes and datasets are available at: `https://github.com/nuochenpku/Orca`.

## 1 Introduction

Conversational Machine Reading Comprehension (CMRC) has aroused an increasing research interest in recent years. Accordingly, lots of large-scale CMRC benchmarks (You et al., 2021; Chen et al., 2023b; Campos et al., 2020; Zaib et al., 2022) have been proposed, such as CoQA (Reddy et al., 2019), QuAc (Choi et al., 2018). Technically, CMRC aims to endow a machine with the ability to understand the given text passage/paragraph and respond appropriately to a set of questions in a conversation. However, despite notable successes in a variety of types and on a sizable scale, there are still a number of important challenges that have rarely been mentioned in prior CMRC efforts:

(1) Conversations in existing CMRC datasets are only grounded on a single/static document passage. Each time a question is asked, it is necessary to consider the passage as the sole source of information. In practice, the natural multi-turn QA is driven by a topic with dynamic knowledge and diverse domains rather than restricted to a static evidence passage. Thus, there is a great discrepancy between the pattern of training data and real application scenarios.

(2) Many annotated answers in these datasets are restricted to span texts in the provided passages. For instance, a significant proportion of CoQA answers are short phrases from the document (Campos et al., 2020). While span texts straightly taken from the original document and basic phrases like "yes/no" might be seen as ground-truth answers, they do not sound like natural responses in real-life. Besides, current Large Language Models (LLMs) are primarily generative in nature. Relying solely on these fixed-length span answers poses limitations in effectively evaluating the quality of LLMs through standard evaluation metrics.

(3) Previous CMRC benchmarks are only built on English corpus, limiting the progress of CMRC in other languages like Chinese. Moreover, questions and knowledge embedded within previous datasets are mostly antiquated, potentially already encompassed within the pre-training corpus of LLMs. As a result, they fail to adequately evaluate the adaptability and understanding abilities of LLMs for new questions and emerging knowledge.

To this end, we collect **Orca**, the first few-shot Chinese CMRC benchmark. Concretely, we hire professional annotators to annotate the following

---

∗`chennuo26@gmail.com`,.Work done when Nuo interned at Xiaoice.AI. † refers to corresponding authors.

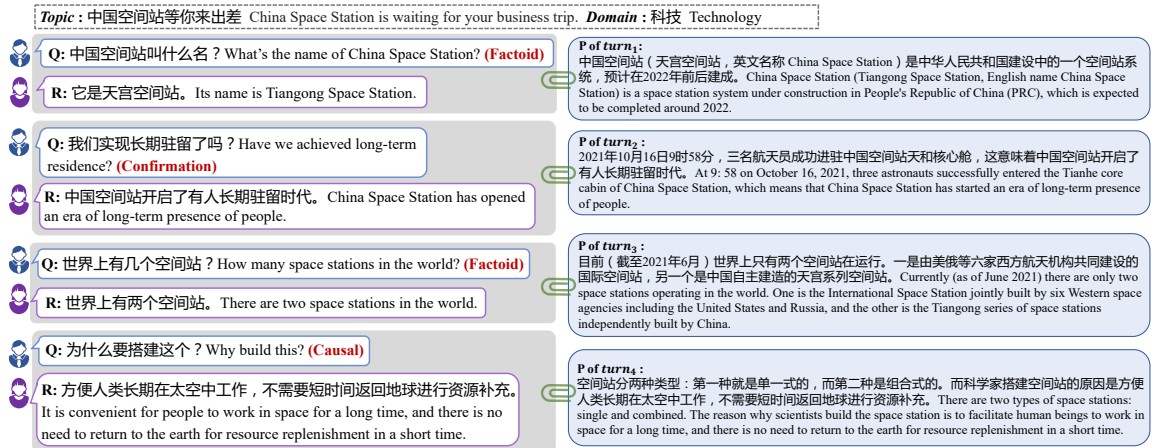

Figure 1: An abridged conversation in **Orca**. **Q**, **R**, and **P** means query, response, and passage respectively.

components in Orca: 1) **Topic**, which consists of several sentences to drive the whole conversation; 2) **Domain**, only one or two words indicate the specific field to which the content of the conversation belongs; 3) **Conversations**, where each turn is assigned a golden knowledgeable passage. Importantly, the question and response in each turn are human-crafted, leading to more coherent conversations. And response-related passages are also manually selected from the search engine. Figure 1 shows an abridged conversation in **Orca**. In total, we annotate 831 conversations and 4742 turns in Orca. We deliberately limit the size of our dataset to build strong CMRC models under few-shot/zero-shot settings, and without the need to collect data for each target domain.

Our dataset has the following salient features: 1) Each turn is assigned with a passage as the knowledge source, i.e., the given passages are dynamically changed rather than only one topic-central passage. Orca gives more in-depth background knowledge about the specific topic in this manner, making the following-up dialogue more natural and diversified. Meanwhile, Orca poses a greater challenge to existing methods since it necessitates modeling both dialogue history and dynamic evidence passages. 2) Answers at each turn are natural and informative responses from human annotation rather than certain spans in the provided passage. With this, we can both evaluate models' comprehension ability and generation ability. 3) Dialogue topics in Orca are collected from November 2021 to November 2022 on Weibo[1], one of the most popular social media platforms in China. This means that the collected data reflects real human inter-

ests, and has never been included in earlier benchmarks. Hence Orca can serve as a comprehensive tool for assessing the adaptability of current LLMs to novel knowledge and emerging questions. Moreover, good results on Orca are of practical interest. 4) We carefully annotate conversations across 33 domains, such as *Society*, *People*, *Celebrity*, *Book*, *Finance*. In contrast, as the most commonly-used datasets, CoQA only has 7 domains and DoQA contains 3 domains. The variety of data domains makes Orca a better testbed to evaluate the generalization of CMRC models. Table 1 shows the comparison between Orca and typical QA datasets.

In this paper, we implement two strong frameworks to tackle the challenge in Orca: 1) In-Context Learning with LLMs like GPT-3 (Brown et al., 2020) and ChatGPT; 2) fine-tuning with language models that are less than 1 billion. In our experiments, the latter includes both end-to-end training and three-stage training. We conduct our experiments under both zero-shot and few-shot settings. The results indicate that Orca provides a challenging evaluation testbed for the CMRC systems.

## 2 Orca

### 2.1 Problem Definition

Generally, the conversational machine reading comprehension task can be illustrated as: Given a conversation at turn t, $<\mathcal{P}, \mathcal{H}_t\ \mathcal{Q}_t, \mathcal{A}_t>$, where $\mathcal{P}$ is the evidence passage, $\mathcal{H}_t$ refers to the dialogue history at turn t, $\mathcal{Q}_t$, $\mathcal{A}_t$ denote the QA pairs at the current turn, respectively. The task requires the model to predict the answer $\mathcal{A}_t$ based on understanding the $\mathcal{P}$, $\mathcal{H}_t$ and $\mathcal{P}$. $\mathcal{A}_t$ often refers to the spans in $\mathcal{P}$.

In contrast, given a conversation at turn t, Orca formalizes the CMRC task as $<\mathcal{T}, \mathcal{P}_t, \mathcal{H}_t\ \mathcal{Q}_t, \mathcal{R}_t>$,

[1]https://m.weibo.cn/

| Dataset | Conversation | Few-shot | Response-related Passage | Answer Type | Language | Domain |
|---|:---:|:---:|:---:|:---:|:---:|:---:|
| SQUAD (2016) | ✗ | ✗ | ✗ | Span | English | - |
| MS.MarCo (2016) | ✗ | ✗ | ✗ | Free-form text | English | - |
| DuReader (2017) | ✗ | ✗ | ✗ | Free-form text | Chinese | - |
| QuAC (2018) | ✔ | ✗ | ✗ | Span | English | - |
| CoQA (2019) | ✔ | ✗ | ✗ | Free-form text | English | 7 |
| DoQA (2020) | ✔ | ✗ | ✗ | Rephrased span | English | 3 |
| **Orca (ours)** | ✔ | ✔ | ✔ | **Natural Response** | **Chinese** | **33** |

Table 1: Comparison of our benchmark with other typical QA datasets. Because Orca is a multi-domain CQA dataset rather than open domain, we don't compare Orca with some open domain benchmarks. We don't report the domain diversity if the original papers don't mention it. Free-form text denotes that the answers include span-text, abstractive text and "yes/no".

where $\mathcal{T}$ is the topic which drives the whole conversation, $\mathcal{P}_t$ represents the evidence passage of turn t, $\mathcal{R}_t$ is the corresponding response. Of note, 1) Since Orca delivers an evidence passage for each round in the conversation, compared to other data sets, it demands the model to have a stronger capacity to comprehend these dynamic knowledge; 2) Each $\mathcal{R}_t$ is the human-annotated natural response rather than $\mathcal{A}_t$, and thus, Orca requires sequence-to-sequence modeling; 3) $\mathcal{H}_t$ in Orca is allowed to include passages of previous turns.

## 2.2 Dataset Collection

As previously stated, a high-quality dataset is required for the development of successful CMRC models. In our dataset, we include the following elements: 1) Topic, a collection of several sentences that serves as the conversation's central focus; 2) Domain where simply one or two words designate the specific field to which the content of the conversation belongs; 3) Conversations, where each turn is allocated a knowledgeable passage. In particular, Orca is collected in three stages: 1) Topic and domain collection; 2) Conversation collection; 3) Quality control. We introduce each of them in the following parts:

**Topic and Domain Collection**   With the goal of collecting data that can reflect real human needs and daily life, we treat the social media platform Weibo as the data source. More than 200 million Chinese users share things around them and talk about hot topics on Weibo every day. We employ ten annotators to find out and write down topics that they are interested in from the hot-topic list [2]. If the selected topic on the list is not a grammatically correct sentence, the marker needs to rewrite a few sentences to re-explain the topic. Meanwhile, the domain of this topic is annotated in Orca. In or-

der to avoid prejudice brought about by individuals, we recruit another annotator to further check the selected topics. We also require the annotators to disregard the topic of the same domain three times in a row, so as to increase the domain diversity. Selected topics in Orca are collected from November 2021 to November 2022 on Weibo, guaranteeing the timeliness and diversity of our data.

**Conversation Collection**   After obtaining topics, we split them into five parts equally. Then, we divide the ten annotators into pairs (one questioner and one responder) and assign them the topics. We ask them to have a conversation around each assigned topic. The questioners are demanded to ask following up questions as many as possible and responders are asked to give natural responses seriously. The conversation begins with the questioner formulating a free-text question from the information on the chosen topic. To ensure that the responses are accurate, comprehensive as well as informative, we let responder use web search engines to find a knowledgeable passage that includes the information of answering this question.

Then, the responder needs to answer the question based on the manually selected passage and their knowledge. Responses are required to be the free-form conversational text, such as a sentence with demonstrative pronouns, making the expression more fluent. To facilitate more natural and consistent conversations, the responder should provide additional information about the answer or the knowledge about the topic to encourage the questioner to continue with questions related to the latest question. Besides, we encourage the questioner to find interesting questions from the recorded dialogues about the topic on Weibo. Dialogues are ended when more than two unanswerable questions were asked or one of the partners decides to end the conversation.

---

[2] https://s.weibo.com/top/summary?cate=realtimehot

| Statistics | | Support | Test | All |
|---|---|---|---|---|
| Conversations | | 200 | 631 | 831 |
| Turns | | 1115 | 3627 | 4742 |
| Turns per Conversations | Max. | 18 | 14 | 18 |
| | Ave. | 5.75 | 5.58 | 5.71 |
| Passage tokens | Max. | 790 | 822 | 822 |
| | Ave. | 92.35 | 90.45 | 91.90 |
| Question tokens | Max. | 19 | 16 | 19 |
| | Ave. | 5.99 | 5.92 | 5.97 |
| Response tokens | Max. | 380 | 197 | 380 |
| | Ave. | 16.31 | 16.50 | 16.35 |

Table 2: Statistics of collected data of **Orca**.

**Quality Control**    To avoid bias and further control the data quality, we have made some strategies to ensure the quality of conversations:

- We employ another five annotators to check whether an inconsistent problem exists in the conversation. Then, annotators are demanded to revise the questions or responses or directly delete the turns that cause inconsistency.

- Simultaneously, gruff questions and responses are revised by annotators, aiming to guarantee natural expressions.

- As our annotation work spans about one year, in order to keep the enthusiasm and work efficiency of annotators, quality check by annotators lasts for about one month. The best-performing annotators will be monetary rewarded, and the worst performers will be re-trained or even separated.

- At last, we will manually re-check the quality to further ensure the high quality of Orca.

All annotators are Chinese native speakers recruited from universities. The annotation costs about 15K US dollars in total, with 2.5 dollars per dialogue turn.

## 3    Dataset Analysis

As a result, Orca contains 831 conversations and 4,742 turns. For each conversation, there are 5.71 turns on average, and each turn is assigned with a knowledgeable passage. We randomly split the collected data into support (train) and test set. Concretely, the support set and test set contain 200 and 631 conversations, separately. Each turn of a conversation needs to be evaluated in experiments. In short, we aim to provide a challenging evaluation

testbed that can evaluate models' ability towards real scenarios in a reasonable way.

### 3.1    Conversation Analysis

As shown in Figure 2 (a), there are 33 different domains in **Orca**, covering a large range of topics in daily life, such as *Society*, *People*, *Celebrity*, *Book*, *Finance*. The variety of domains makes **Orca** more challenging, which can effectively evaluate model's generalization ability towards different domains.

To evaluate model's generalization ability towards various domains, we extract 200 conversations from **Orca** as the support set and treat the rest data as test set. Table 2 shows the statistics of **Orca**. Each conversation contains 5.71 turns on average, which is closer to the number of QA pairs in a human conversation (Campos et al., 2020). There are about 91.9 tokens per passage and 16.35 tokens per response on average, indicating the abundant information of manually retrieved passages.

In terms of the link between a question and its conversation history, we categorize questions as dependent or independent of the conversation history. According to our findings, only 19.9% of questions do not need coreference with the conversational history and can be answered on their own. Explicit coreference indicators such as he, she, and it appears in over half of the queries (57.8%). These are either entities, people, buildings or events mentioned in the conversation. The remaining 22.3% lack explicit coreference indicators but are indirectly related to an item or event, like simple following-up questions "Why this?". The analysis shows Orca poses challenging linguistic phenomena to build effective CMRC models.

### 3.2    Query Analysis

Query types in Orca can be classified into 5 streams: *Factoid* (39.1%), *Causal* (30.1%), *List* (13.0%), *Confirmation* (15.4%) and *Hypothetical* (2.4%). Different from existing datasets (Choi et al., 2018; Reddy et al., 2019; Christmann et al., 2019; Saha et al., 2018; Chen et al., 2021b) in which most queries are *Factoid*, query types in **Orca** are more diverse. For example, factoid questions accounted for about 60% of CoQA. In contrast, 60.9% of the questions are non-factoid questions on Orca, where *Causal*, *List*, and *Confirmation* all account for a considerable part.

Figure 2 (b) shows the distribution of the bigram prefixes of queries. We observe that most of the first word of queries are similar to other datasets,

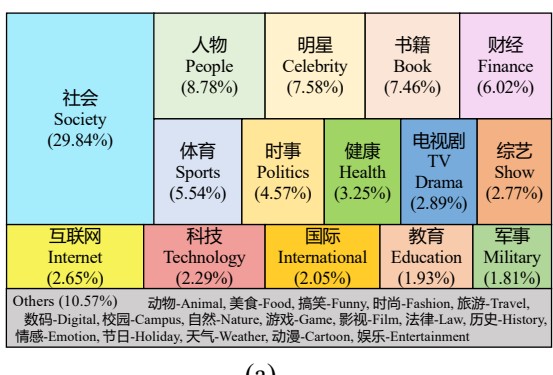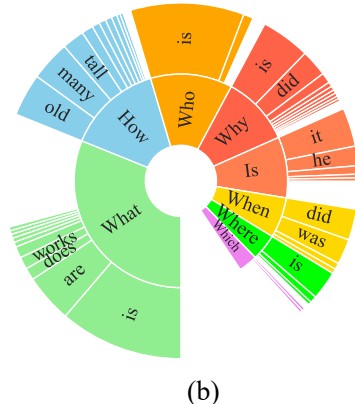

(a) (b)

Figure 2: (a) Distribution of domains in **Orca**. (b) Distribution of the bigram prefixes of querys in **Orca**. For ease of reading, English translation is only provided here.

like QuAC, CoQA and CSQA. However, queries beginning with "Why" and "Is" account for a larger proportion compared with other datasets. We find that "Why" pattern is mainly related to queries with *Causal* type (eg. Why is he popular in China?) and "Is" pattern is mainly related to queries with *Confirmation* type (eg. Is it a developed country?). The diversity and relatively uniform distribution of queries in Orca provide conversation scenarios that are as realistic as possible.

### 3.3 Response Analysis

According to the statistics of all responses in Orca, we observe that **69.30%** of responses cannot be found as exact text spans from the passage. Only **30.70%** of responses have overlap spans in the given passage (like entities). It indicates that most of the responses in Orca are free-form expressions that are more colloquial, calling for models with strong generative ability.

## 4 Models

In this section, we introduce three strong baselines to address the challenge in Orca: 1) In-context learning with large language models; 2) fine-tuning including end-to-end training and three-stage training which consists of three modules: Query Rewriter, Passage Reader, and Response Rewriter (short for QPR).

### 4.1 In-context Learning

Previous works have proved that directly utilizing LLMs with in-context learning could lead to promising performances in zero-shot. In-context learning means the learning procedure that the models learn to make predictions in a target task via conditioning on a few input label pairs In our experiments, we employ GPT-3

(text-davinci-002), ChatGLLM-130B and ChatGPT (gpt3.5-turbo) as the backbone. Concretely, given the task descriptions as prompts on the train set, we validate the performances of these LLMs on the test set from Orca.

### 4.2 End-to-End

As most of the responses in Orca are free-form texts, rather than text spans extracted from the given passages, we build our end-to-end pipeline with several widely-used generative models: T5 and Bart. **T5** (Raffel et al., 2020) is a pre-trained encoder-decoder Transformer model which has achieved strong results on a diverse set of benchmarks, such as question answering. We use a pre-trained T5 based on Chinese (Wang et al., 2022) with 784 million parameters for experiments. **BART** (Lewis et al., 2020) is a pre-trained encoder-decoder Transformer model for text generation tasks which has 375 million parameters.

In our implementation, we optimize the model with the widely used negative log-likelihood loss during training. At turn t in a conversation, the optimized objective can be formulated as:

$$
\begin{aligned}
\mathcal{L} &= -\log(p_\theta(\mathcal{R}_t \mid \mathcal{T}, \mathcal{P}_t, \mathcal{H}_t, \mathcal{Q}_t)) \\
&= -\sum_{i=1}^{|\mathcal{R}_t|} log(p_\theta(r_i \mid \mathcal{T}, \mathcal{P}_t, \mathcal{H}_t, \mathcal{Q}_t)).
\end{aligned} \tag{1}
$$

### 4.3 QPR Framework

As explained in Section 1, human cognition process towards CMRC can be split into three steps. To this end, given the related passage of each turn, we further design a QPR that consists of Query Rewriter, Passage Reader, and Response Rewriter. In this architecture, given a question at turn t, Query Rewriter module first rewrites the question $\mathcal{Q}_t$ to a

| Shots (Sessions) | Method | BLEU-1 | BLEU-2 | Distinct-1 | Distinct-2 | ROUGE-L | EM |
|---|---|---|---|---|---|---|---|
| | *In-Context Learning* | | | | | | |
| 0-shot | ChatGLM-130B | 12.96 | 9.42 | 2.21 | 31.37 | 23.71 | 1.05 |
| | GPT-3 w/o P | 25.41 | 18.81 | 5.27 | 44.84 | 34.02 | 2.04 |
| | GPT-3 | 32.29 | 29.05 | 3.86 | 44.77 | 36.77 | 3.25 |
| | ChatGPT w/o P | 23.84 | 17.37 | 2.68 | 32.87 | 31.22 | 1.38 |
| | ChatGPT | 35.93 | 33.21 | 2.46 | 38.13 | 47.72 | 3.03 |
| 1-shot | GPT-3 | 33.88 | 32.59 | 3.59 | 40.67 | 39.22 | 1.01 |
| | ChatGPT | 36.70 | 32.81 | 3.13 | 43.23 | 47.87 | 3.35 |
| 5-shot | GPT-3 | 34.56 | 33.40 | 3.05 | 39.63 | 41.01 | 3.41 |
| | ChatGPT | 41.11 | 37.81 | 4.31 | 46.44 | 57.96 | 5.03 |
| | *Fine-Tuning* | | | | | | |
| — | QPR | 61.10 | 58.21 | 4.47 | 48.51 | 62.07 | 14.83 |
| 5-shot | BART | 24.72 | 21.37 | 3.46 | 39.72 | 26.79 | 1.54 |
| | T5 | 42.41 | 39.70 | 3.65 | 41.95 | 47.39 | 6.89 |
| 10-shot | BART | 38.73 | 35.30 | 4.24 | 44.51 | 40.87 | 5.82 |
| | T5 | 46.59 | 43.98 | 3.58 | 43.09 | 57.80 | 12.54 |
| 200-shot (full data) | BART | 61.59 | 59.05 | **5.34** | **51.56** | 69.90 | 25.23 |
| | T5 | **68.25** | **66.11** | 4.50 | 46.82 | **73.06** | **31.40** |

Table 3: Comparison of baselines on **Orca**, which have significant improvement with $p$-value $< 0.05$. "GPT-3 w/o P" means we use the conversations without passages to evaluate GPT-3.

more natural text $\hat{\mathcal{Q}}_t$. Then Passage Reader module is responsible for generating answers $\mathcal{A}_t$ based on $(\mathcal{T}, \mathcal{P}_t, \mathcal{H}_t, \hat{\mathcal{Q}}_t)$. At last, Response Rewriter module takes $\mathcal{A}_t$ as inputs and generates the final predicted response. Considering the fact that Orca lacks labels of re-written questions, we post-train these modules on some benchmarks, and then test the resulting models in zero-shot setting on Orca. The following presents our implementation details.

**Query Rewriter.** Demonstrative pronouns and abridged expressions often appear in natural conversations. To comprehend queries better, we use the dataset from Su et al. (2019) to fine-tune a Query Rewriter module based on BART, realizing anaphora resolution and omission completion.

**Passage Reader.** Given an answer-related passage and the query after Query Rewriter, we need to generate an answer from the passage. We use the dataset from He et al. (2018) to fine-tune a generative Passage Reader based on BART, obtaining a preliminary answer.

**Response Rewriter.** To further improve the generated answer and make the expression more natural, we feed the answer and original query without Query Rewriter into Response Rewriter to get the final response. We use the dataset from Chen et al. (2023a) to fine-tune the generative Response Rewriter based on BART.

## 5 Experiments

In this section, we first introduce experimental settings. Then automatic and human evaluation metrics are introduced to evaluate model performances. At last, we present extensive experimental results.

### 5.1 Experimental Setting

We implement experiments under zero-shot/few-shot settings to evaluate model's performance towards unseen domains with few samples. Following existing works (Mi et al., 2022), Orca provides **multiple few-shot splits** to measure variance in CMRC models across various splits. Concretely, we split 5 sessions and 10 sessions from the support set and define them as 5-shot and 10-shot, respectively. 0-shot means nonuse of support set. In a word, experiments are implemented under 0-session, 5-session, 10-session and 200-session settings (named as 0-shot, 5-shot, 10-shot and 200-shot respectively). For QPR framework, due to the deficiency of ground-truth of Query Rewriter and Passage Reader, it cannot be fine-tuned based on the support set. Therefore, we only evaluate this system under 0-shot setting. Furthermore, we test LLMs in 0-shot and few-shot settings with in-context learning. Due to the significant expense of fine-tuning it and the max input sequence limit, we use text-davinci-003-16k as backbone of ChatGPT. T5 and BART are evaluated under all the settings. We present training details in Appendix A.

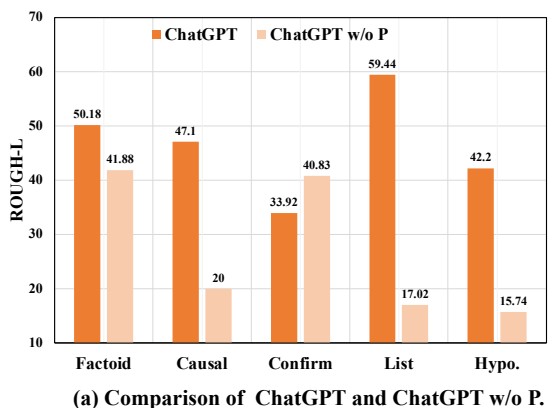

(a) Comparison of ChatGPT and ChatGPT w/o P.

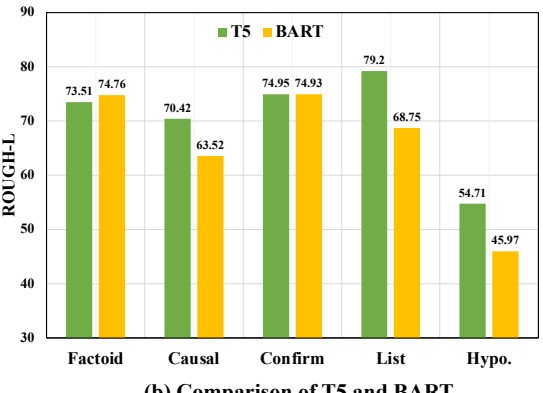

(b) Comparison of T5 and BART.

Figure 3: Automatic evaluation results of different methods under various query types.

| Models | Relev. | Comple. | Nature. |
|---|---|---|---|
| *0-shot* | | | |
| ChatGPT | 2.48 | 2.50 | 3.08 |
| QPR System | **3.00** | **3.04** | **3.30** |
| *200-shot* | | | |
| BART | 2.92 | 3.80 | **3.45** |
| T5 | **3.12** | **3.82** | 3.36 |

Table 4: Human evaluation results (%), which have significant improvement with $p$-value $< 0.05$.

## 5.2 Evaluation Metrics

**Automatic Metrics.** We employ the following widely-used metrics to validate the lexical, semantic, and diversity aspects of the generated responses, including exact matching score (**EM**), **ROUGE-L** (Lin, 2004), **BLEU-1**, **BLEU-2** (Papineni et al., 2002), **Distinct-1**, and **Distinct-2** (Li et al., 2016). **Human Evaluation.** In order to further get a full view of how well the model works, we recruit 3 annotators to rate the generated response of the best two models under 0-shot setting and 200-shot setting. Annotators are asked to give three scores according to **Relevance**, **Completeness**, and **Naturality** with levels of {1, 3, 5}, which represent unreadable, mediocre and flawless, respectively. We present the details of scoring rules in the Appendix. For all evaluation metrics, the larger number denotes better performances. We present annotation details in Table 6.

## 5.3 Results

**Automatic Evaluations** Table 3 reports our baselines' results of in-context learning and fine-tuning. From the table, we can observe that: (1) Under in-context learning settings, ChatGPT performs best in terms of all metrics when compared with

| Topic | An earthquake occurred last night in Aba, Sichuan, and lasted all night |
|---|---|
| **Domain** | Society |
| **Conversation History** | |
| **Question** | When did this earthquake occur? |
| **Response** | It happened on June 10 at 00:03 am. |
| **......** | |
| **Current Trun** | |
| **Question** | What is the maximum magnitude of it? |
| **Passage** | Malcom City, Aba Prefecture, Sichuan, June 10, 0:03 a.m., a 5.8 magnitude earthquake, followed by several earthquakes ...... including 9 in Malcom City and 1 in Hongyuan County, with a maximum magnitude of **6.0**. |
| **Response** | **The maximum magnitude of this earthquake is 6.0.** |
| **T5** | It is at 5.8 level. |
| **BART** | It is 5.8. |

Table 5: An error case of BART and T5 under 200-shot settings. We highlight the ground-truth response or the evidence text in the table.

GPT-3 and CharGLM-130B. (2) Although we don't fine-tune QPR systems in Orca, it still outperforms ChatGPT under 0-shot settings. We argue that two reasons may contribute to its performance: 1) The architecture of QPR is more consistent with the human cognition process towards CMRC; 2) post-training QPR on several similar Chinese benchmarks brings more benefit. (3) With the increase of training samples for fine-tuning, model performances on the test set become better. Comparing ROUGE-L scores under 5-shot and 10-shot settings, the improvement of BART and T5 is **12.25%** on average. Learning from only 10 support samples, their performances improve markedly and can catch up with that of the QPR system. Given 200 support samples, end-to-end baselines achieve sig-

nificant promotions and surpass the QPR system.

From the above observations, we can conclude that: (1) The disparity between the performance of LLMs under the in-context learning setup and the results obtained through fine-tuning smaller language models is striking. Particularly noteworthy is the fact that, despite the substantial difference in parameter count, with T5 being more than two hundred times smaller than GPT-3, fine-tuning T5 with a mere five dialogue sessions yields superior outcomes compared to GPT-3. These findings serve as a compelling testament to both the difficulty of Orca and the untapped potential of LLMs in tackling the CMRC task. Moreover, through our analysis of the results, we have observed that ChatGLM and ChatGPT tend to exhibit a tendency to refuse responses or provide excessively verbose and convoluted answers, which primarily contributes to their lower performances in automated metrics. (2) Compared with GPT-3, "GPT-3 w/o P", ChatGPT and "ChatGPT w/o P" sheds light that **Orca** contains plenty of conversations with fresh knowledge and it's hard for GPT-3 and ChatGPT to generate accurate responses only based on its parameters pre-trained by old knowledge. It also proves the value of passages provided by Orca.

**Human Evaluations**  To comprehensively assess the performance of these models, we randomly sampled 100 dialogue sessions from the test set for manual evaluation. In the table below, we present the results for ChatGPT, QPR, as well as BART and T5 fine-tuned on the full data. T5 and BART demonstrate comparable performance across the three perspectives evaluated but exhibit more pronounced effects compared to the other two models, proving their good ability to understand the passage, history and question, so as to generate complete and natural responses. Based on the analysis of generated responses, we find that Chat-GPT tends to generate long responses with extra information, ignoring the response completeness sometimes. The human evaluation results basically match their performances in terms of automatic evaluations.

In summary, the results under 0-shot and few-shot settings prove the effectiveness of the proposed three pipelines, but they still remain the challenge in Orca. For instance, none of the models performs above 80% on automatic metrics. From the human evaluation perspective, they achieve the worst results on **Relev.**, showing they still suffer from deficiencies in modeling.

### 5.4 Analysis

We further explore the model performances on different query types. Similarly, we conduct experiments on zero-shot and 200-shot settings.

**Influence of Query Type**  Figure 3 (a) reports the results of different types of queries for fine-grained analysis. First and foremost, it is evident that Chat-GPT demonstrates remarkable proficiency in addressing list-type questions and performs worst in confirm-type questions. Moreover, the utilization of knowledge notably enhances ChatGPT's capabilities, particularly in the domains of causal and list-based question answering. Focusing on 200-shot setting, Figure 3 (b) shows the results of T5 and BART under different query types. T5 almost outperforms BART under all query types, depending on its advantage in the number of model parameters. We obverse that BART shows a poor performance on *Causal* queries, *List* queries, and *Hypothetical* queries in particular. According to Figure 2, these three query types account for **45.5%** in **Orca**, proving that the variety of query types poses a thorny challenge for CMRC research. Moreover, a horizontal comparison of figure (a) and (b) reveals a distinct advantage of T5 and BART over ChatGPT in answering confirmation, causal, and factoid-type questions after fine-tuning.

**Error Analysis of BART and T5**  We conduct an error case study to show the challenge of Orca. As seen in Table 5, we first find that there is a coreference phenomenon in the latest question. That is, utilizing *it* to indicate *the earthquake* in the above question. Besides, the collected passage doesn't mention the highest earthquake magnitude clearly; instead, it states the average level, which confuses the model. As a result, two models are generated with *5.8* rather than the ground-truth answer *6.0*. This example demonstrates that not only do the questions in Orca encompass a wide range of linguistic phenomena, but it is also challenging to comprehend the gathered passages. More case studies are in Appendix.

**Error Analysis of ChatGPT and GPT-3**  From the table 3, it is evident that even in 1-shot and 5-shot scenarios, the performance of ChatGPT and GPT-3 is not as strong as expected, and it falls significantly behind fine-tuned T5 models. Regarding the specific reasons for the performance of Chat-

GPT and GPT-3, our case analysis points to the following factors:

1. **Output in English**: In specific cases, despite repeated resampling, ChatGPT tends to generate responses in English. We speculate that this behavior might be influenced by the presence of certain English words unique to those contexts in the preceding text.

2. **Declining to answer**: In scenarios requiring new knowledge or information, even when ample relevant information has been provided to answer the questions. Based on the knowledge up until 2021, ChatGPT sometimes refuses to respond, citing incomplete information as its rationale. Here we provide an example: This dialogue is about a Chinese internet celebrity from 2022, Brother Man from Peking University. The current round of questions is about how Brother Man makes a living, and the provided passage already details the way he earns money. However, ChatGPT still refuses to answer the question, citing a lack of relevant information. We present one detailed example in Appendix F, Table 8.

3. **Closed-domain hallucination**: We have observed instances where ChatGPT exhibits a closed-domain illusion, mistakenly assuming limitations to its knowledge domain. This can result in responses that appear overly cautious or confined to past information. Concretely, hallucination can be defined as: 1) **Intrinsic hallucination**: the generated response is contradictory to the dialogue history or the external knowledge sentences. 2) **Extrinsic hallucination**: the generated response is hard to verify with the dialogue history or the external knowledge sentences. We present two detailed examples in Appendix F, Table 8.

## 6 Conclusion

We propose the first few-shot benchmark Orca for Chinese CMRC. Unlike existing CMRC datasets that each dialog contains a static passage, we assign each turn a golden response-related passage in Orca, which is more consistent with real scenarios. All the conversations in Orca are hot-topic driven and collected from various domains, responses are natural and informative via human annotation. In this way, model's comprehension ability and generation ability towards real scenarios can be evaluated

reasonably. Moreover, zero-shot and few-shot settings are provided to evaluate model's learning ability towards unseen domains with few samples. Experimental results of our proposed three baselines indicate that Orca provides a challenging testbed for CMRC research.

## Limitations

The main target of this paper is towards developing and testing Chinese strong CMRC models. To this end, we collect and propose Orca. More generally, we expect the core contribution of this paper can be seen as a universal benchmark for testing LLMs' ability to new questions and knowledge under CMRC task. Admittedly, the data in the proposed dataset from the Harry Potter Series is relatively small and restricted to Chinese communities. These concerns warrant further research and consideration when utilizing this work to build intelligent CMRC systems in the virtual world.

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

## A  Training details

We fine-tune BART for 50 epochs with a learning rate of 2e-5. The batch size is set to 10. The max length of input and output sequences are set to 512 and 128, respectively. For T5, we fine-tune it for 20 epochs with the initial learning rate of 1e-4 and batch size of 8. The max length of input and output sequences are set to 512 and 300, respectively. In this work, we utilize large version of BART[3] and base version of T5[4] and *text-davinci-002* version of GPT-3 in our experiments.

## B  Related Work

More and more MRC datasets have been proposed to promote the development of QA systems (Chen et al., 2021a; Song et al., 2023; Ishii et al., 2022; Chen et al., 2022a,b; Ye et al., 2023). Tabel 1 presents the comparison of text-based MRC datasets in recent years. The release of QuAC (Choi et al., 2018) and CoQA (Reddy et al., 2019) have aroused great interest in CMRC. In these datasets, each conversation is assigned a document passage. Questioner asks questions based on the passage and the answerer finds evidence answers from it. However, this pattern is quite different from natural human-human conversations. In reality, a natural conversation is driven by a certain topic and an evidence passage that is available for each turn doesn't exist (You et al., 2022). Limiting the conversation content into a given document passage is inconsistent with real scenarios. Thus, model's evaluation on these datasets could not reflect its ability towards realistic scenarios reasonably. Besides, almost CMRC datasets are collected from English corpus, limiting the language diversity in CMRC tasks.

Unlike existing datasets, we propose a Chinese CMRC benchmark with various hot topics and each turn of a dialog is assigned a golden response-related passage. Thus, models' comprehension ability and generation ability towards real scenarios can be evaluated more reasonably. Furthermore, we collect Orca in few-shot settings, challenging models to learn unseen domains with few samples. Although one can question previous full-data CMRC datasets also could support few-shot training for CMRC models, we argue that it could lead to unclear comparisons due to the inconsistent settings of different works on these datasets (Chen et al., 2022b). In contrast, we present a single standard benchmark for thorough comparisons.

---

[3] https://github.com/fastnlp/CPT
[4] https://github.com/ZhuiyiTechnology/t5-pegasus

| Aspect | Score | Rule |
|--------|-------|------|
| **Rel.** | 1 | The prediction is unrelated to query or topic. |
| | 3 | The prediction doesn't cover the semantics of golden response accurately. It only covers a part of information mentioned by golden response or involve some redundant information. |
| | 5 | The prediction covers the semantics of golden response accurately. |
| **Com.** | 1 | The expression of the prediction is uncompleted. |
| | 3 | The expression of the prediction is completed but it's a bit awkward to read it with the query. |
| | 5 | The expression of the prediction is completed and smooth. |
| **Nat.** | 1 | The expression of the prediction is unnatural. |
| | 3 | The expression of the prediction is somewhat natural but it's slightly insufficient with the absence of demonstrative pronoun. |
| | 5 | The expression of the prediction is natural and has demonstrative pronoun. |

Table 6: The details of scoring rules of human evaluation.

## C  Details of Human Evaluation

For human evaluation, we ask annotators to give three scores according to Relevance, Completeness, and Naturality with level of {0, 1, 2}, respectively. Table 6 shows the details of scoring rules of the three aspects.

## D  Case Study of GhatGPT with and without Passage under 0-shot Setting

As shown in Table 3, we evaluate GhatGPT with and without passage under 0-shot setting. "Ghat-GPT w/o P" performs worse on most of the automatic metrics. For intuitive analysis, we present cases of GhatGPT w/ and w/o passage in Table 7. It indicates that GhatGPT cannot generate accurate response without passage in some cases, specially the queries that need fresh knowledge. The reason is that GhatGPT pre-trained on old knowledge is probably unaware of new knowledge. Moreover, the comparison of GhatGPT with and without passage proves the value of golden passages provided by **Orca**.

## E  Case Study of T5 under 200-shot Setting

As shown in Table 9, we present two cases of T5 under 200-shot setting. Table 9(a) shows a case that T5 generated response is the same as golden response. It indicates that T5 has strong learning ability and has an excellent performance under 200-shot setting. Table 9(b) shows a case that T5 generated response is slightly worse than golden response. Compared with the golden response, T5's response has obvious information loss. The rea-

son could be that the related passage is too long to generate an accurate response from it.

## F  Case studies of ChatGPT

We present one example of declining to respond and two hallucination examples of ChatGPT in Table 8. Here, we give more detailed explanations of hallucination examples.

In the first example, the conversation history revolves around inquiring about some achievements of Ai Fukuhara. However, there is an explicit topic shift in the current part of the conversation. It starts to ask about who Zhang Jike is and whether he has won Olympic championships. The passage we provided contains ample information to accurately answer this question. However, the response generated by ChatGPT is entirely inaccurate:

- Firstly, a significant portion of its response remains entangled in the previous discussion about Ai Fukuhara, leading to an answer that is off-topic and redundant.

- Secondly, in its response about Zhang Jike, there is a glaring inconsistency. It initially states that Zhang Jike has won the 2012 Olympic championship but later concludes by stating that he has not won an Olympic championship. This incorrect response showcases both a failure to address the current question appropriately and internal inconsistencies in the information provided.

In the second example, the current turn's question in the conversation pertains to how cancer is detected. The provided passage also offers detailed steps for cancer detection. However, **ChatGPT's response veers off track by revisiting the previous conversation history, discussing the definition of cancer, its causes, symptoms, and related matters**. This response is off-topic and fails to directly address the inquiry about how cancer is detected, as outlined in the current question.

| Topic: | 中国空间站等你来出差 |
| --- | --- |
| | *China Space Station is waiting for your business trip.* |
| Domain: | 科技 (*Technology*) |

| Query: | 中国空间站叫什么名？(*Factoid*) |
| --- | --- |
| | What's the name of China Space Station? |
| Passage: | 中国空间站（天宫空间站，英文名称China Space Station）是中华人民共和国建设中的一个空间站系统... |
| | China Space Station (Tiangong Space Station, English name China Space Station) is a space station system under construction in People's Republic of China... |
| Response: | 它是天宫空间站。 |
| | Its name is **Tiangong Space Station**. |
| **GPT-3 w/o P**: | 它叫做天宫二号。 |
| | It's called **Tiangong II**. |
| **GPT-3 w/ P**: | 天宫空间站，英文名称China Space Station。 |
| | **Tiangong Space Station**. Its English name is China Space Station. |

...

| Query: | 它可以长期驻留多少个人？(*Factoid*) |
| --- | --- |
| | How many people can stay there for a long time? |
| Passage: | 空间站轨道高度为400 450公里，倾角42 43度，设计寿命为10年，长期驻留3人，总重量可达180吨，以进行较大规模的空间应用... |
| | The orbital height of the space station is 400-450 kilometers, the inclination is 42-43 degrees, the design life is 10 years, there are 3 long-term residents, and the total weight can reach 180 tons for large-scale space applications... |
| Response: | 它可以长期驻留3人。 |
| | **Three people** can stay there for a long time. |
| **GhatGPT w/o P**: | 它可以长期驻留6个人。 |
| | **Six people** can stay there for a long time. |
| **GhatGPT w/ P**: | 它可以长期驻留3人。 |
| | **Three people** can stay there for a long time. |

...

| Query: | 世界上有几个空间站？(*Factoid*) |
| --- | --- |
| | How many space stations in the world? |
| Passage: | 目前（截至2021年6月）世界上只有两个空间站在运行。一是由美俄等六家西方航天机构共同建设的国际空间站，另一个是中国自主建造的天宫系列空间站。 |
| | Currently (as of June 2021) there are only two space stations operating in the world. One is the International Space Station jointly built by six western space agencies including the United States and Russia, and the other is the Tiangong series of space stations independently built by China. |
| Response: | 世界上有2个空间站。 |
| | There are **two space stations** in the world. |
| **GhatGPT w/o P**: | 目前世界上共有5个空间站。 |
| | Currently, there are **five space stations** in the world. |
| **GhatGPT w/ P**: | 目前（截至2021年6月）世界上只有两个空间站在运行。一是由美俄等六家西方航天机构共同建设的国际空间站，另一个是中国自主建造的天宫系列空间站。 |
| | Currently (as of June 2021) there are only **two space stations** operating in the world. One is the International Space Station jointly built by six western space agencies including the United States and Russia, and the other is the Tiangong series of space stations independently built by China. |

Table 7: Cases of GhatGPT with and without passage under 0-shot setting.

| | Example of Declining to answer |
|---|---|
| **Conversation History** | Who is Brother Man? Brother Man from Peking University was born in the countryside and grew up in the northeast of China. His sister studied at Tsinghua University, while he attended Peking University and is engaged in online education in the field of the internet. Did he complete his undergraduate studies at Peking University? Yes, Brother Man graduated from Peking University. |
| **Question** | How does he earn money? |
| **Passage** | Brother Man earns money through live streaming online education. Brother Man from Peking University gained popularity mainly by sharing his journey of getting into Peking University. Born in the countryside and raised in the northeast, with his sister also getting into Tsinghua University, Brother Man's story made me understand a truth: life is far more challenging than one might imagine, but you are also much stronger than you think. |
| **Generated Response** | Based on the knowledge up until 2021, I can't answer this question because I don't know who "Brother Man" you mentioned is. |

| | Example-1 of Closed-domain hallucination |
|---|---|
| **Conversation History** | What nationality is Ai Fukuhara? Ai Fukuhara is of Japanese nationality. Why is she so popular in China? It's quite simple – Ai Fukuhara has spent a significant amount of time in China. Moreover, her fluent Mandarin with a northeastern accent eliminates any sense of distance. Beyond her ability to speak Chinese, her amiable personality and high emotional intelligence are more crucial factors. Plus, she's known to be a bit of a crybaby. |
| **Question** | Do you know Zhang Jike? Has he won the Olympic Men's Singles Championship? |
| **Passage** | Zhang Jike is a Chinese male table tennis player. In 2012, he won the World Table Tennis Team Championships and secured the Men's Singles gold medal at the London Olympics in August, completing a grand slam of individual championships that includes the World Championships, World Cup, and Olympics. |
| **Generated Response** | Ai Fukuhara is of Japanese nationality and her popularity in China can be attributed to several reasons: She has spent a significant amount of time in China, establishing a deeper connection with the Chinese people. She is fluent in Mandarin with a northeastern accent, which eliminates any sense of distance between her and the Chinese population. Her good nature and high emotional intelligence contribute to her likability. Her tendency to shed tears easily adds to her perceived charm and cuteness. In 2012, she won the World Table Tennis Team Championships and secured the Women's Singles gold medal at the 2012 London Olympics, achieving a grand slam of individual championships that includes the World Championships, World Cup, and Olympics. **Zhang Jike has not won an Olympic Men's Singles gold medal.** |

| | Example-2 of Closed-domain hallucination |
|---|---|
| **Conversation History** | What is the essence of cancer? Cancer is a process where normal cells in the human body undergo changes, transforming from normal cells to atypical cells. What are the causes of this disease? They can be categorized into exogenous factors and endogenous factors. The symptoms of this disease include: Systemic symptoms; Local symptoms of the cancer; Symptoms caused by distant metastasis of the cancer... What is chemotherapy? Chemotherapy is short for chemical therapy. It involves using chemical drugs to kill cancer cells for therapeutic purposes. How can this disease be prevented? Maintain a regular daily routine; Adopt healthy dietary habits; Avoid smoking. Other aspects: Patients should actively engage in prevention, early detection, and early treatment. |
| **Question** | How is this disease diagnosed? |
| **Passage** | Different types of cancers require different types of tests. Commonly used tests include blood tests, imaging tests, and invasive tests. Here are the details: Imaging tests: In clinical practice, if an abnormality is found in the lungs during a chest X-ray, a high-resolution CT scan of the chest is usually performed to assess the local situation. It's also recommended to conduct a CT scan of the upper abdomen and the entire abdomen. This not only helps determine the size of the lung tumor but also identifies any potential invasion of the adrenal glands and liver. Additionally, a whole-body bone scan or cranial MRI might be necessary for a comprehensive assessment. Blood tests: Blood tests provide information about the basic functional status of the body, including complete blood count, liver function, kidney function, coagulation function, etc. Tumor markers can be detected through blood tests. Invasive tests: For example, in the case of lung cancer, a fiberoptic bronchoscopy or a lung biopsy might be performed. If there's suspicion of digestive tract tumors, an endoscopy of the stomach or colon might be necessary. Enlarged lymph nodes in the neck might require a lymph node biopsy or even excision for examination. It's important to note that the specific tests conducted would depend on the suspected type and location of the cancer, as well as the individual patient's condition. |
| **Generated Response** | Cancer is a malignant tumor originating from transformed cells that grow uncontrollably. It can result from various factors, including external influences like radiation, chemicals, and internal factors such as genetics. Symptoms vary by cancer type but include systemic effects, local changes, and metastasis-related issues. Some cancers can be cured, depending on type and timing. After successful treatment, patients can live long, influenced by multiple factors. Treatment involves surgery, chemotherapy, radiation, immunotherapy, and more, tailored to cancer specifics and patient health. Follow-up care is crucial for maintaining health post-treatment. |

Table 8: Error cases of ChatGPT.

| | |
|---|---|
| **Topic**: | 年轻人为啥不使用手机壳了？ |
| | *Why don't young people use mobile phone cases*? |
| **Domain**: | 社会 (*Society*) |

| | |
|---|---|
| Query: | 手机壳有哪些作用？ |
| | What are the functions of mobile phone case? |
| Passage: | 手机保护壳不仅作为装饰品，更能保护手机... |
| | It is not only used as an ornament, but also can protect the mobile phone... |
| Response: | 手机保护壳不仅作为装饰品，更能保护手机更能保护手机，防摔、防刮、防水和防震。 |
| | It is not only used as an ornament, but also can protect the mobile phone, and it can prevent falling, scratching, water and shock. |
| ... | |
| **Query**: | 使用手机壳有什么坏处？ (***Hypothetical***) |
| | What are the disadvantages of using a mobile phone case? |
| **Passage**: | 手机最好不用手机壳，原因如下：1，手机壳会影响手机散热，导致手机寿命降低... |
| | It's best not to use a mobile phone case for the following reasons: 1. The mobile phone case will affect the heat dissipation of the mobile phone, which will reduce the life of the mobile phone... |
| **Response**: | 1. 手机壳会影响手机散热，导致手机寿命降低。2. 手机壳影响手机的美观，让手机失去人体学的外形。3. 手机壳的制作材料多为化学合成材料，经常使用不利于身体健康。 |
| | 1. The phone case will affect the heat dissipation of the phone, resulting in a reduction in the life of the phone. 2. The phone case affects the beauty of the phone and makes the phone lose its ergonomic appearance. 3. Mobile phone cases are mostly made of chemical synthetic materials, and frequent use is not conducive to health. |
| **T5**: | 1. 手机壳会影响手机散热，导致手机寿命降低。2. 手机壳影响手机的美观，让手机失去人体学的外形。3. 手机壳的制作材料多为化学合成材料，经常使用不利于身体健康。 |
| | 1. The phone case will affect the heat dissipation of the phone, resulting in a reduction in the life of the phone. 2. The phone case affects the beauty of the phone and makes the phone lose its ergonomic appearance. 3. Mobile phone cases are mostly made of chemical synthetic materials, and frequent use is not conducive to health. |

(a) An example that T5 performs well compared with golden response.

| | |
|---|---|
| **Topic**: | 南京地铁公安迎来两只超小号警犬 |
| | *Two small police dogs are welcomed by Nanjing Metro Police.* |
| **Domain**: | 社会 (*Society*) |

| | |
|---|---|
| Query: | 成为警犬需要具备哪些条件？ |
| | What are the requirements to become a police dog? |
| Passage: | 警犬需要具备的条件是很强的视觉、听觉能力；极强的嗅觉能力... |
| | Police dogs need strong visual and auditory abilities; strong sense of smell... |
| Response: | 警犬需要具备很强的视觉、听觉能力... |
| | Police dogs need strong visual and auditory abilities... |
| ... | |
| **Query**: | 它退役后去哪？ (***Causal***) |
| | Where does it go after retirement? |
| **Passage**: | 军犬在退役后大部分仍会留在军营... |
| | Most of them will remain in the barracks after retirement... |
| **Response**: | 它们在退役后大部分仍会留在军营，安置在"军犬疗养院"中，由专门负责照顾它们的训导员悉心照料，安享晚年。还有部分退役警犬会被领养走。 |
| | Most of the them will remain in the barracks after retirement and be placed in the special sanatorium. They will be carefully taken care of by the trainers and enjoy their old age. **Some retired police dogs will be adopted**. |
| **T5**: | 它们会留在军营，安置在"军犬疗养院"中，由专门负责照顾它们的训导员悉心照料，安享晚年。 |
| | Most of the them will remain in the barracks after retirement and be placed in the special sanatorium. They will be carefully taken care of by the trainers and enjoy their old age. |

(b) An example that T5 shows slightly worse performance compared with golden response.

Table 9: Two cases of T5 under 200-shot setting.