# OpenReview forum: "Orca: A Few-shot Benchmark for Chinese Conversational Machine Reading Comprehension"
_EMNLP/2023/Conference — EMNLP 2023 Findings_

### Official Review · Reviewer_fQ1z · 2023-07-24

**Soundness:** 3

**Excitement:**

3: Ambivalent: It has merits (e.g., it reports state-of-the-art results, the idea is nice), but there are key weaknesses (e.g., it describes incremental work), and it can significantly benefit from another round of revision. However, I won't object to accepting it if my co-reviewers champion it.

**Paper Topic And Main Contributions:**

The paper has proposed the first Chinese Conversational Machine Reading Comprehension benchmark Orca. The authors have demonstrated the quality and usefulness of the Orca dataset by evaluating several state-of-the-art models on the benchmark. The paper is well-written and presents a clear and comprehensive description of the Orca dataset.

**Reasons To Accept:**

The paper presents a valuable contribution to the field of natural language processing by proposing a dataset for the Chinese Conversational Machine Reading Comprehension task. In addition to proposing the dataset, the authors have provided a comprehensive evaluation of several state-of-the-art models based on LLM and PLM methods.

**Reasons To Reject:**

It is surprising that ChatGPT, when given an answer passage, could not generate a correct answer, as shown by the low BLEU score in Table 3. However, it is important to consider whether the BLEU score accurately reflects answer correctness in this context. While the BLEU score is suitable for evaluating text span, it may not be the most appropriate metric for evaluating natural text generated by LLMs in conversational settings. It is possible that ChatGPT may provide a correct answer but not in the exact format of the golden answer. Therefore, it would be valuable to consider additional metrics to more accurately reflect the answer's correctness. In addition, it would be valuable to analyze the poor performance of ChatGPT in generating correct answers given an answer passage. Specifically, it would be interesting to investigate whether the poor performance is due to a poor understanding of Chinese or a poor retrieval ability to search for correct answer evidence. I am willing to raise the scores if my consideration could be solved.

**Reproducibility:**

4: Could mostly reproduce the results, but there may be some variation because of sample variance or minor variations in their interpretation of the protocol or method.

**Reviewer Confidence:**

3: Pretty sure, but there's a chance I missed something. Although I have a good feel for this area in general, I did not carefully check the paper's details, e.g., the math, experimental design, or novelty.

---

> ### Author Rebuttal · Authors · 2023-08-28
>
> We thank you for your meticulous review and invaluable suggestions on our work. We would like to address and clarify the key issues and suggestions you raised:
>
> **Q1**: Question about using BLEU scores to evaluate  LLMs such as BLEU and Distinct.
>
> >**A1**:  In response to your concern regarding the use of metrics such as BLEU and Distinct in our paper, I appreciate the opportunity to provide a thorough explanation of our choices as authors.
>
> We acknowledge the concern about using metrics like BLEU and Distinct in evaluating the generated texts of LLMs. ***While it's true these metrics have limitations in capturing the full quality of LLMs, there is currently no universally accepted automatic metric that offers a complete evaluation of large model-generated text.*** While BLEU and Distinct might not fully capture the quality of text generated by large models, they still provide valuable insights into certain aspects of the model outputs. In addition to the BELU and Distinct metrics you mentioned, we have also employed the ROUGEL and EM metrics, which focus on fuzzy matching and exact matching, respectively. We aim to evaluate the model's performance from multiple metric perspectives.
>
> Moreover, to further corroborate our automatic evaluation results, we conduct evaluations based on three human evaluated metrics, as shown in Table 4. The results demonstrate a positive consistency between automatic and human evaluations. We look forward to future research that advances the field's understanding of evaluation and provides more accurate and comprehensive metrics for assessing the true quality of these models' outputs.
>
> **Q2**: Analyze the poor performances of ChatGPT
>
> >**A2**:  Thanks for your thoughtful concern about the poor performance of ChatGPT.
>
> In our analysis, the suboptimal performance of ChatGPT can be mainly attributed to several reasons as outlined below:
>
> - **Output in English**: In specific cases, despite repeated resampling, ChatGPT tends to generate responses in English. We speculate that this behavior might be influenced by the presence of certain English words unique to those contexts in the preceding text.
> - **Declining to answer**: In scenarios requiring new knowledge or information, even when ample relevant information has been provided to answer the questions. Based on the knowledge up until 2021, ChatGPT sometimes refuses to respond, citing incomplete information as its rationale. Here we provide an example: This dialogue is about a Chinese internet celebrity from 2022, Brother Man from Peking University. The current round of questions is about how Brother Man makes a living, and the provided passage already details the way he earns money. However, ChatGPT still refuses to answer the question, citing a lack of relevant information.
>
> |   |                  Example of Declining to answer         |
> | ------------------------------ | ------------------------- |
> | Dialogue History               | Who is Brother Man? Brother Man from Peking University was born in the countryside and grew up in the northeast of China. His sister studied at Tsinghua University, while he attended Peking University and is engaged in online education in the field of the internet. Did he complete his undergraduate studies at Peking University? Yes, Brother Man graduated from Peking University. |
> | Current Question               | How does he earn money?    |
> | Current Passage                | Brother Man earns money through live streaming online education. Brother Man from Peking University gained popularity mainly by sharing his journey of getting into Peking University. Born in the countryside and raised in the northeast, with his sister also getting into Tsinghua University, Brother Man's story made me understand a truth: life is far more challenging than one might imagine, but you are also much stronger than you think. |
> | Generated Response                       | **Based on the knowledge up until 2021, I can't answer this question because I don't know who "Brother Man" you mentioned is.** |
>
> - **Closed-domain hallucination**: We have observed instances where ChatGPT exhibits a closed-domain illusion [A][B], mistakenly assuming limitations to its knowledge domain. This can result in responses that appear overly cautious or confined to past information. Concretely, hallucination can be defined as:
>     - **Intrinsic hallucination**: the generated response is contradictory to the dialogue history or the external knowledge sentences.
>     - **Extrinsic hallucination**: the generated response is hard to verify with the dialogue history or the external knowledge sentences.
>
> Responses with extrinsic hallucination are impossible to verify with the given inputs.  We provide two illustrative examples below to further elaborate on this point.
>
> |  |Example-1 of Closed-domain hallucination|
> | ----------- | -----------|
> |Dialogue History|What nationality is Ai Fukuhara? Ai Fukuhara is of Japanese nationality. Why is she so popular in China? It's quite simple – Ai Fukuhara has spent a significant amount of time in China. Moreover, her fluent Mandarin with a northeastern accent eliminates any sense of distance. Beyond her ability to speak Chinese, her amiable personality and high emotional intelligence are more crucial factors. Plus, she's known to be a bit of a crybaby.  |
> |Current Question|Do you know Zhang Jike? Has he won the Olympic Men's Singles Championship? |
> |Current Passage|Zhang Jike is a Chinese male table tennis player. In 2012, he won the World Table Tennis Team Championships and secured the Men's Singles gold medal at the London Olympics in August, completing a grand slam of individual championships that includes the World Championships, World Cup, and Olympics. |
> |Generated Response|Ai Fukuhara is of Japanese nationality and her popularity in China can be attributed to several reasons: She has spent a significant amount of time in China, establishing a deeper connection with the Chinese people. She is fluent in Mandarin with a northeastern accent, which eliminates any sense of distance between her and the Chinese population. Her good nature and high emotional intelligence contribute to her likability. Her tendency to shed tears easily adds to her perceived charm and cuteness. In 2012, she won the World Table Tennis Team Championships and secured the Women's Singles gold medal at the 2012 London Olympics, achieving a grand slam of individual championships that includes the World Championships, World Cup, and Olympics. **Zhang Jike has not won an Olympic Men's Singles gold medal.**|
>
> **In the first example**, the conversation history revolves around inquiring about some achievements of Ai Fukuhara. However, there is an explicit topic shift in the current part of the conversation. It starts to ask about who Zhang Jike is and whether he has won Olympic championships. The passage we provided contains ample information to accurately answer this question.
>
> However, the response generated by ChatGPT is entirely inaccurate:
> - Firstly, a significant portion of its response remains entangled in the previous discussion about Ai Fukuhara, leading to an answer that is off-topic and redundant.
> - Secondly, in its response about Zhang Jike, there is a glaring inconsistency. It initially states that Zhang Jike has won the 2012 Olympic championship but later concludes by stating that he has not won an Olympic championship.
> This incorrect response showcases both a failure to address the current question appropriately and internal inconsistencies in the information provided.
>
>
> | |      Example-2 of Closed-domain hallucination                                                                           |
> |------------|-------------------------------------------------------------------------------------|
> | Dialogue History | What is the essence of cancer? Cancer is a process where normal cells in the human body undergo changes, transforming from normal cells to atypical cells. What are the causes of this disease? They can be categorized into exogenous factors and endogenous factors. The symptoms of this disease include: Systemic symptoms; Local symptoms of the cancer; Symptoms caused by distant metastasis of the cancer... What is chemotherapy? Chemotherapy is short for chemical therapy. It involves using chemical drugs to kill cancer cells for therapeutic purposes. How can this disease be prevented? Maintain a regular daily routine; Adopt healthy dietary habits; Avoid smoking. Other aspects: Patients should actively engage in prevention, early detection, and early treatment. |
> | Current Question | How is this disease diagnosed? |
> | Current Passage | Different types of cancers require different types of tests. Commonly used tests include blood tests, imaging tests, and invasive tests. Here are the details: Imaging tests: In clinical practice, if an abnormality is found in the lungs during a chest X-ray, a high-resolution CT scan of the chest is usually performed to assess the local situation. It's also recommended to conduct a CT scan of the upper abdomen and the entire abdomen. This not only helps determine the size of the lung tumor but also identifies any potential invasion of the adrenal glands and liver. Additionally, a whole-body bone scan or cranial MRI might be necessary for a comprehensive assessment. Blood tests: Blood tests provide information about the basic functional status of the body, including complete blood count, liver function, kidney function, coagulation function, etc. Tumor markers can also be detected through blood tests. Invasive tests: For example, in the case of lung cancer, a fiberoptic bronchoscopy or a lung biopsy might be performed. If there's suspicion of digestive tract tumors, an endoscopy of the stomach or colon might be necessary. Enlarged lymph nodes in the neck might require a lymph node biopsy or even excision for examination. It's important to note that the specific tests conducted would depend on the suspected type and location of the cancer, as well as the individual patient's condition. |
> | Generated Response | Cancer is a malignant tumor originating from transformed cells that grow uncontrollably. It can result from various factors, including external influences like radiation, chemicals, and internal factors such as genetics. Symptoms vary by cancer type but include systemic effects, local changes, and metastasis-related issues. Some cancers can be cured, depending on type and timing. After successful treatment, patients can live long, influenced by multiple factors. Treatment involves surgery, chemotherapy, radiation, immunotherapy, and more, tailored to cancer specifics and patient health. Follow-up care is crucial for maintaining health post-treatment. |
>
> **In the second example**, the current turn's question in the conversation pertains to how cancer is detected. The provided passage also offers detailed steps for cancer detection. However, **ChatGPT's response veers off track by revisiting the previous conversation history, discussing the definition of cancer, its causes, symptoms, and related matters**. This response is off-topic and fails to directly address the inquiry about how cancer is detected, as outlined in the current question.
>
> This instance demonstrates a lack of relevant and focused response, where ChatGPT misses the mark by not providing a direct answer to the question regarding cancer detection procedures.
>
> We hope our response can address your concern, and we will add these analyses and cases to our Appendix.
>
> [A]: Survey of Hallucination in Natural Language Generation
>
> [B]: Performance and exploration of ChatGPT in medical examination, records and education in Chinese: Pave the way for medical AI

---

### Official Review · Reviewer_vj1q · 2023-08-04

**Typos Grammar Style And Presentation Improvements:** line 342
**Soundness:** 3

**Excitement:**

3: Ambivalent: It has merits (e.g., it reports state-of-the-art results, the idea is nice), but there are key weaknesses (e.g., it describes incremental work), and it can significantly benefit from another round of revision. However, I won't object to accepting it if my co-reviewers champion it.

**Paper Topic And Main Contributions:**

The paper presents a new dataset for conversational question answering. Different from previous work one dialogue only involves one passage, the proposed new benchmark considers multiple passages that one dialogue can involve. In each round, there is a passage as the reference.



**Questions For The Authors:**

1. In Table 3, how do you address the issue that the ChatGPT's response is open-ended and sometimes, it may generate some unrelated content to the answer, e.g, "I am happy to help you.", "As an AI language model ..". How these contents can affect your evaluation.

**Reasons To Accept:**

1. A new benchmark: The paper provides a new benchmark for studying conversational question answering.

The new benchmark considers a new issue that is ignored by previous datasets, that is each question may involve different passages, instead of just one.

The new benchmark should be useful to the community.

**Reasons To Reject:**

There are some issues for the proposed new benchmark:

1. Evaluation Metric: the new benchmark emphasize the gold answer for each question a natural sentence instead of phrases that can be extracted from the passage. However, the evaluation metric used for evaluation is BELU and Distinct.

Such evaluation metrics can not well measure the real performance of models, since BELU only measures the token overlaps between the candidate and reference. For the baseline of ChatGPT, under the zero-shot setting, its response is quite open-ended, so it is very hard to measure ChatGPT's response.

So, I would think the evaluation metric for this benchmark is not good.

2. I have the doubt for the challenge of the new benchmark.

With the birth of ChatGPT / GPT-4 / Claude, their conversation ability is quite good. For conversational QA, it should be quite easy for them to address, given the external passage for each question.

**Reproducibility:**

4: Could mostly reproduce the results, but there may be some variation because of sample variance or minor variations in their interpretation of the protocol or method.

**Reviewer Confidence:**

4: Quite sure. I tried to check the important points carefully. It's unlikely, though conceivable, that I missed something that should affect my ratings.

---

> ### Author Rebuttal · Authors · 2023-08-28
>
> Firstly, sorry for the typos and we sincerely thank you for dedicating your time and effort to review our paper and provide valuable feedback. In response to the concerns raised, we would like to offer the following clarifications:
>
> **Q1**: Question about evaluation metrics such as BLEU and Distinct.
>
> **A1**: In response to your concern regarding the use of metrics such as BLEU and Distinct in our paper, I appreciate the opportunity to provide a thorough explanation of our choices as authors.
>
> We acknowledge the concern about using metrics like BLEU and Distinct in evaluating the generated texts of LLMs. While it's true these metrics have limitations in capturing the full quality of LLMs, **there is currently no universally accepted automatic metric that offers a complete evaluation of LLM-generated texts**. While BLEU and Distinct might not fully capture the quality of texts generated by LLMs, they still provide valuable insights into certain aspects of the model outputs such as diversity. In addition to the BELU and Distinct metrics you mentioned, we have also employed the ROUGEL and EM metrics, which focus on fuzzy matching and exact matching, respectively. We aim to evaluate the model's performance from multiple metric perspectives.
>
> Moreover, to further corroborate our automatic evaluation results, we conduct evaluations based on ***three human-constructed metrics***, as shown in Table 4. The results demonstrate a positive consistency between automatic and human evaluations. We look forward to future research that advances the field's understanding of evaluation and provides more accurate and comprehensive metrics for assessing the true quality of these models' outputs.
>
> **Q2**: Question about the challenge of Orca: With the birth of ChatGPT / GPT-4 / Claude, their conversation ability is quite good. For conversational QA, it should be quite easy for them to address, given the external passage for each question.
>
> >**A2**: Thanks for your concerns about the difficulty of Orca, we have several points to clarify:
>
> - Our dataset, based on the CMRC framework, provides dynamic context passages for each question within the conversation. Specifically, at each turn of a dialogue, the model's input of in-context learning could include the task description, the conversation's topic, domain, dialogue history, knowledge from previous rounds, the current question, and corresponding contextual knowledge. As conversations deepen, the number of turns and the length of input gradually increase. Our statistical analysis (shown in the table below) reveals that nearly 1/4 of the dataset has input lengths exceeding 2K tokens, with even 10% of instances surpassing 4K tokens. This presents three significant challenges for LLMs:
>
>     - First, testing their memory and comprehension abilities for longer texts;
>     - Second, demanding effective strategies for modeling lengthy texts when input lengths exceed model limits. Simple in-context learning may not suffice to handle our dataset, as argued in [C].
>    - In Orca, the model's comprehension task is not limited to a simple question-external passage pair. It involves grasping extensive dialogue context, dynamic knowledge updates, and potential topic shifts.
>
> These challenges continue to remain pertinent and unsolved for current LLMs.
>
> | Input Length |0-500 | 500-2k | 2k-4k |  >4k|
> | :----------- | :-----------: | :-----------: | :-----------: | :-----------: |
> |Percentage (%)| 34.3% | 41.9% | 14.2% | 9.6% |
>
>
> - Recent studies [A] [B] have pointed out the existence of "closed-domain hallucination" in current LLMs, where erroneous answers are generated despite the presence of external passages:
>     -  **Intrinsic hallucination**: the generated response is contradictory to the dialogue history or the external knowledge sentences.
>     -  **Extrinsic hallucination**: the generated response is hard to verify with the dialogue history or the external knowledge sentences. Responses with extrinsic hallucination are impossible to verify with the given inputs. “Pickwick hotel” might be “in san diego”, and Djokovic may have been “in the top ten singles players of the world”. However, we do not have enough information to check the truth of these statements. Our dataset, being novel and diverse, increases the probability of such hallucinations, further complicating the evaluation.
>
> | |Example (We omit provided passages of previous turns) |
> |-----------|:--------------------------------------------------|
> | Dialogue History |  What nationality is Ai Fukuhara? Ai Fukuhara is of Japanese nationality. Why is she so popular in China? It's quite simple – Ai Fukuhara has spent a significant amount of time in China. Moreover, her fluent Mandarin with a northeastern accent eliminates any sense of distance. Beyond her ability to speak Chinese, her amiable personality and high emotional intelligence are more crucial factors. Plus, she's known to be a bit of a crybaby. |
> | Current Question |  Do you know Zhang Jike? Has he won the Olympic Men's Singles Championship? |
> | Current Passage |  Zhang Jike is a Chinese male table tennis player. In 2012, he won the World Table Tennis Team Championships and secured the Men's Singles gold medal at the London Olympics in August, completing a grand slam of individual championships that includes the World Championships, World Cup, and Olympics. |
> | Generated Response |  Ai Fukuhara is of Japanese nationality and her popularity in China can be attributed to several reasons: She has spent a significant amount of time in China, establishing a deeper connection with the Chinese people.  In 2012, she won the World Table Tennis Team Championships and secured the Women's Singles gold medal at the 2012 London Olympics, achieving a grand slam of individual championships that includes the World Championships, World Cup, and Olympics. **Zhang Jike has not won an Olympic Men's Singles gold medal.** |
>
>  Here, we present a hallucination case of ChatGPT in Orca. In this example, the conversation history revolves around inquiring about some achievements of Ai Fukuhara. However, there is an explicit topic shift in the current part of the conversation. It starts to ask about who Zhang Jike is and whether he has won Olympic championships. The passage we provided contains ample information to accurately answer this question. ***However, the response generated by ChatGPT is entirely inaccurate***:
>
> - Firstly, a significant portion of its response remains entangled in the previous discussion about Ai Fukuhara, leading to an answer that is off-topic and redundant.
> - Secondly, in its response about Zhang Jike, there is a glaring inconsistency. It initially states that Zhang Jike has won the 2012 Olympic championship but later concludes by stating that he has not won an Olympic championship. This incorrect response showcases both a failure to address the current question appropriately and internal inconsistencies in the information provided.
>
> ***Taken together, the challenges posed by Orca go beyond the simplicity of question-external passage pairs, extending into the intricacies of dialogue context, dynamic knowledge incorporation, limited max token lengths, and handling topic shifts.*** These challenges remain significant hurdles for current state-of-the-art LLMs and underscore the necessity of addressing them for more effective CMRC.
>
> [A]: Survey of Hallucination in Natural Language Generation
>
> [B]: Performance and exploration of ChatGPT in medical examination, records and education in Chinese: Pave the way for medical AI
>
> [C]: MemoryBank: Enhancing Large Language Models with Long-Term Memory
>
> **Q3**: Question about how to evaluate ChatGPT's performances when its response is open-ended sometimes.
>
> >**A3**: Thank you for raising concerns regarding the open-ended responses from ChatGPT. In our experiments, our prompts specifically request the model to avoid producing irrelevant statements, such as "as a language model" or "as an AI." Additionally, we apply post-processing to the outputs, utilizing techniques such as regular expression matching, to remove statements that are unrelated to the answers. The final outputs from our experiment show no presence of such unrelated expressions.
>
> **Q4**: Typo: "bel pairs In our experiments" -> "bel pairs in our experiments"
>
> >**A4**: Sorry for the typos, we will revise this mistake in our final version.

---

### Official Review · Reviewer_e1ov · 2023-08-08

**Soundness:** 3

**Excitement:**

3: Ambivalent: It has merits (e.g., it reports state-of-the-art results, the idea is nice), but there are key weaknesses (e.g., it describes incremental work), and it can significantly benefit from another round of revision. However, I won't object to accepting it if my co-reviewers champion it.

**Paper Topic And Main Contributions:**

This paper presents a new Chinese language dataset for machine reasoning comprehension that focuses on conversational QA based on multiple documents. The dataset is constructed with (crowdsourced) human interactions focusing on trendy topics, amounting to 831 conversations and 4742 interactions in total. The paper also presents baseline performances of ChatGPT and others, demonstrating much room for improvements.


**Reasons To Accept:**

- a new corpus of Chinese language conversational reading comprehension

- compared to the prior resources that assumed the exact phrase from the provided paragraph as answers, this paper aims to support more naturalistic and free-form answers

- compared to the prior resources that assumed one external document to ground the conversation on, this paper aims to support multiple documents to ground the conversation

**Reasons To Reject:**

- the experiments for baselines, especially ChatGPT could've been executed better. As is, only zero-shot performance is reported for ChatGPT, while the authors could've included few-shot cases as well.

- Moreover, due to the particularity about how the external document is provided primarily by the responder interleaved in between the conversation turns, how exactly the instruction was phrased for ChatGPT will be very important. I wish the authors experimented with different strategies carefully and reported the exact prompts/instructions used in the experiments in the appendix for completeness and reproducibility.

- I have a mixed feeling about the quality of the gold label of Table 5 example (and the likely consequences of such noise in the annotation on the baseline performances). it seems debatable to me that the gold answer provided is necessarily the only correct answer (or the most naturalistic answer), especially considering that the preceding interaction was about the earthquake that happened on Jun 10 at 00:03am, combined with the fact that the retrieved document itself characterized that the earthquake as "a 5.8 magnitude earthquake" explicitly. If the annotations had such noise, it is likely that only supervised methods that are capable of learning the particularities of the annotations can perform well, which will inherently influence the perceived gap between supervised approaches and ChatGPT performance. While I would be excited to see a failure mode of ChatGPT, this particular annotation just doesn't seem quite right to me

- the role of retrieved documents in the evaluation seems rather unclear. Currently it seems the evaluation assumes that the external documents are provided by an oracle, but in reality I imagine those documents need to be automatically retrieved...? (thus it might make sense to include such evaluation settings as well...?)

**Reproducibility:**

3: Could reproduce the results with some difficulty. The settings of parameters are underspecified or subjectively determined; the training/evaluation data are not widely available.

**Reviewer Confidence:**

4: Quite sure. I tried to check the important points carefully. It's unlikely, though conceivable, that I missed something that should affect my ratings.

---

> ### Author Rebuttal · Authors · 2023-08-28
>
> First and foremost, we appreciate the time and effort you dedicated to reviewing our paper and providing valuable feedback. In response to the concerns raised by the reviewer, we offer the following rebuttals.
>
> **Q1**: Add Few-shot experiments of ChatGPT.
>
> >**A1**: Thanks for your thoughtful advice. When constructing the dataset, we create support sets of 5, 10, and 200 shots. Apart from the zero-shot experiments we report, we indeed attempt few-shot trials, such as the 5-shot, which means we add training samples from our 5-shot support set as exemplars in our prompts.  However, given that some of our test samples involved lengthy dialogue histories and passages, sometimes spanning several dozen turns, ***they exceed the maximum token limit of ChatGPT, rendering it incapable of generating a relevant answer***. Given the unfairness of this comparison to ChatGPT under such circumstances, we thus decide not to report these results in the paper.
>
> **Q2**: Add  Experiments with different strategies carefully and report the exact prompts/instructions used in the experiments in the appendix for completeness and reproducibility.
>
> >**A2**:  Thanks for your thoughtful advice. In fact, during our experimentation process, we have already investigated the impact of different prompts on the model's performance. However, due to space constraints, we only report the optimal results in the paper. As shown in the two tables below, we present the various prompts we utilize, along with their corresponding results based on ChatGPT. Just as you suggest, we will include these discussions in our Appendix.
>
>  Here, we only provide English translation of our prompts.
>
> | Prompt |Content |
> | :----------- | :-----------|
> |Prompt-1|  The following is a conversation between two users, User A and User B. Your task is to reply to the latest question from User A based on their question and the conversation context. Additional relevant information has been provided to assist you in answering User A's question. Please use the provided knowledge and the conversation history to formulate your response. Please remember you just need to directly give your response and avoid producing irrelevant statements.|
> |Prompt-2| You are a professional  dialogue agent. Below, there will be a conversation about a certain topic involving the user and the assistant. Your task is to respond to the user's latest question based on  the conversation content. To assist you in answering questions, we also provide some relevant knowledge. You need to incorporate the provided knowledge along with the conversation history to formulate your response. Please remember you just need to directly give your response and avoid producing irrelevant statements.|
> |Prompt-3| The conversation involves two users, User A and User B. Your role is to reply a response to the most recent question from User A.Your task is to respond to the latest round of questions from User A from the perspective of User B, based on User A's questions and the conversation context. To assist you in answering User A's question, we have provided some relevant information. You should incorporate the provided knowledge along with the conversation history to give your response. Please remember you just need to directly give your response and avoid producing irrelevant statements. |
>
> | ChatGPT |BLEU-1 |BLEU-2 |Distinct-1 |Distinct-2 |ROUGE-L |EM|
> | :----------- | :-----------: | :----------- | :-----------: | :----------- | :-----------: |:-----------: |
> | Prompt-1 |35.30 |32.90 |2.59 |39.00 |46.14|1.91|
> | Prompt-2 |34.45 |32.60 |3.45 |39.63 |45.67 |3.01|
> | Prompt-3 |35.93 |33.21 |2.46 |38.13 |47.22 |3.03|
>
> **Q3**: Question about quality of the gold label of Table 5 example
>
> >**A3**: Sorry for the misunderstanding.  After conducting a thorough review of our dataset, we've identified an error in Table 5. The question should be "What was the maximum earthquake magnitude?" instead of "What was the earthquake magnitude?" This is a typo that we will rectify in the upcoming version of our paper.
>
> **Q4**: the role of retrieved documents in the evaluation seems rather unclear. Currently it seems the evaluation assumes that the external documents are provided by an oracle, but in reality I imagine those documents need to be automatically retrieved...? (thus it might make sense to include such evaluation settings as well...?)
>
> >**A4**: Thank you for your thoughtful inquiry. However, there appears to be a minor misunderstanding in your interpretation. As we've illustrated in both the introduction and problem formulation sections of our paper, we focus on the ***CMRC (Conversational Machine Reading Comprehension)*** task, as opposed to the conventional CQA (Conversational Question Answering) task. It's noteworthy that the Orca dataset also aligns with the CMRC category.
>
> The primary differentiation between these tasks lies in the fact that CQA tasks don't furnish predefined passages or documents as repositories of knowledge. Consequently, models engaged in CQA tasks are mandated to perform document retrieval during conversations in order to procure pertinent information. On the other hand, ****CMRC tasks involve provided passages and conversations, requiring models to understand passage content in conversation****. This difference means Orca doesn't require a document retrieval step, eliminating the need to assess the precision and quality of retrieval documents.

---

### Meta-Review · Area_Chair_1uBa · 2023-09-18

**Recommendation:** 3

**Metareview:**

This paper contributes a conversational machine reading comprehension dataset in Chinese. The performance of models, both commercial (in zero-shot and few shot ICL setting) and open (few and many shot finetuned) was evaluated on the dataset, showing room for improvement. The reviewers all believe (and I agree) that the resource is valuable, and that the task setting is realistic and challenging: conversations grounded in multiple documents, and answers not necessarily being spans in the input contexts. The authors provided additional results and analysis to some of the concerns raised by the reviewers, particularly prompt sensitivity and error analysis of ChatGPT on this dataset.

---

### Decision · Program_Chairs · 2023-10-07

**Decision:**

Accept-Findings

**Comment:**

This paper contributes a conversational machine reading comprehension dataset in Chinese. The performance of models, both commercial (in zero-shot and few shot ICL setting) and open (few and many shot finetuned) was evaluated on the dataset, showing room for improvement. The reviewers all believe (and I agree) that the resource is valuable, and that the task setting is realistic and challenging: conversations grounded in multiple documents, and answers not necessarily being spans in the input contexts. The authors provided additional results and analysis to some of the concerns raised by the reviewers, particularly prompt sensitivity and error analysis of ChatGPT on this dataset.